# Constrained Exploitability Descent: Finding Mixed-Strategy Nash Equilibrium by Offline Reinforcement Learning

## Abstract

This paper presents Constrained Exploitability Descent (CED), a novel model-free offline reinforcement learning algorithm for solving adversarial Markov games. CED is a game-theoretic approach combined with policy constraint methods from offline RL. While policy constraints can perturb the optimal pure-strategy solutions in single-agent scenarios, we find this side effect can be mitigated when it comes to solving adversarial games, where the optimal policy can be a mixed-strategy Nash equilibrium. We theoretically prove that, under the uniform coverage assumption on the dataset, CED converges to a stationary point in deterministic two-player zero-sum Markov games. The min-player policy at the stationary point satisfies the necessary condition for making up an exact mixed-strategy Nash equilibrium, even when the offline dataset is fixed and finite. Compared to the model-based method of Exploitability Descent that optimizes the max-player policy, our convergence result no longer relies on the generalized gradient. Experiments in matrix games, a tree-form game, and an infinite-horizon soccer game verify that a single run of CED leads to an optimal min-player policy when the practical offline data guarantees uniform coverage. Besides, CED achieves significantly lower NashConv compared to an existing pessimism-based method and can gradually improve the behavior policy even under non-uniform coverage.

## 1 Introduction

Offline reinforcement learning (RL) (Levine et al., 2020) has become an increasingly attractive research topic in recent years since data-driven learning of policies is appealing, especially in scenarios where the interaction with the environment is expensive, e.g., robotic manipulation, autonomous driving, and health care. Offline RL faces an inherent challenge of distributional shift (Ross et al., 2011), which arises from visiting out-of-distribution states and actions. A direct way to address this issue is to apply policy constraints, which bound distributional shift by constraining how much the learned policy differs from the behavior policy (Kakade & Langford, 2002; Schulman et al., 2015). In single-agent Markov decision processes (MDPs), such constraints can lead to suboptimality of the learned policy since the optimal policy is usually a pure strategy, which assigns the optimal action probability one at each state (Sutton & Barto, 2018). Since the behavior policy derived from a set of offline transitions can hardly be a pure strategy, applying policy constraints with respect to the behavior policy will sacrifice the optimality of the learned policy, even if the coverage of the offline data is theoretically sufficient for learning the optimal policy (e.g., satisfying uniform concentration).

For multi-agent scenarios, the optimal solution can still be a pure strategy when it is fully cooperative. However, in adversarial games, e.g., two-player zero-sum Markov games (MGs), we usually characterize the optimal solution with the concept of Nash equilibrium (NE), which admits mixed strategies. For example, in a two-player Rock-Paper-Scissors (RPS) game, the unique NE is the mixed strategy $\left(\frac{1}{3}, \frac{1}{3}, \frac{1}{3}\right)$ for both players. It is thus possible that policy constraint methods under a mixed-strategy behavior policy may not sacrifice policy optimality in MGs. While recent research in the field of game theory has developed various efficient equilibrium-learning dynamics that can be extended into model-free RL algorithms (Lanctot et al., 2017; Lockhart et al., 2019), it has not been examined if these algorithms can be further combined with existing offline learning techniques (Siegel et al.,

2020; Wu et al., 2019) while still guaranteeing to learn an exact Nash equilibrium under sufficient assumptions on the data coverage.

On the other hand, while the existing pessimism-based methods are provably efficient for solving offline MDPs and MGs (see Jin et al. (2021); Xiong et al. (2023)), they have certain limitations when they are practically applied to real-world games. First, while these methods can be near-optimal when the environment contains uncertainty, they still require infinitely many samples to fully capture the stochasticity on the game and achieve the optimal solution, i.e., Nash equilibrium. However, when the game is deterministic (e.g., chess and Go), they become no longer optimal since the game transition can be determined by a finite number of samples, which are already sufficient for finding NE. Second, existing pessimism-based methods usually require the information about game horizon (see Cui & Du (2022a;b); Zhong et al. (2022); Xiong et al. (2023)) or dynamics model (see Yan et al. (2024)) to solve Markov games. Zhang et al. (2023), as an exception, suffers from the problem of computational inefficiency. Therefore, it is still challenging to propose a completely model-free method that is capable of solving infinite-horizon MGs offline and, at the same time, does not lose theoretical guarantee and computational efficiency.

With the above-mentioned concerns, we try to answer the following question:

*Is it possible to find mixed-strategy Nash equilibrium offline for adversarial games using model-free equilibrium-learning dynamics with policy constraints?*

This paper provides a positive answer to this question. Specifically, the contributions are threefold:

- We propose a novel model-free RL algorithm for finding mixed-strategy Nash equilibrium in adversarial Markov games from a finite offline dataset. The algorithm, named Constrained Exploitability Descent (CED), is constructed by extending the ideas of policy constraint methods from offline RL and a game theoretic approach, Exploitability Descent (ED).

- We prove that, under the uniform coverage assumption, CED converges in deterministic two-player zero-sum MGs (Theorem 1) without relying on a generalized gradient like ED. We further show that the min-player policy becomes unexploitable when the opponent converges to an interior point of the constrained policy space (Theorem 2). By exchanging the status of the two players and running CED twice, we can obtain a potential mixed-strategy NE.

- We verify the equilibrium-finding capability of CED by conducting experiments in matrix games, a tree-form game, and a soccer game. Given a dataset with uniform coverage, CED can find NE policies in all scenarios, with the practical NashConv significantly lower than the baseline derived from a pessimism-based method. As an offline RL algorithm, CED also gradually improves the behavior policy under non-uniform coverage of offline game data.

## 2 RELATED WORK

**Pessimism-based methods in offline games.** The recent works that directly examine offline games basically focus on sample complexity and rely on pessimistic value functions, which have been well understood in single-agent RL (Rashidinejad et al., 2021; Xie et al., 2021). These works typically append bonuses to the original Bellman operators and obtain confidence bounds on the duality gap for the policy computed from dynamic programming (Cui & Du, 2022a;b; Zhong et al., 2022; Xiong et al., 2023; Yan et al., 2024). In the theoretical analyses, corresponding concentration inequality is utilized to capture the stochasticity of the transition function. As a fundamental work, Cui & Du (2022a) proves that the coverage assumption of unilateral concentration is sufficient for finding Nash equilibrium offline in two-player zero-sum games by providing algorithms with Hoeffding/Bernstein-type bonuses. Subsequent works improve the sample complexity (see Cui & Du (2022b)) and extend the analyses to more complex scenarios concerning linear/general function approximations (see Xiong et al. (2023); Zhang et al. (2023)).

**Equilibrium-learning dynamics.** The field of algorithmic game theory (Roughgarden, 2016; Nisan et al., 2007) examines a wide range of equilibrium-learning dynamics. While the basic method of dynamic programming (or more simply, backward induction) can only deal with perfect information games like Markov games, game-theoretic learning dynamics, including Fictitous Play (FP) (Brown, 1951), Policy Space Response Oracle (PSRO) (Lanctot et al., 2017), and Exploitability Descent

(ED) (Lockhart et al., 2019), can solve a broad class of games even with imperfect information. Among them, PSRO is already extended through deep reinforcement learning. ED exhibits last-iterate convergence and is conducive to offline RL extensions. While other methods like optimistic multiplicative weights update (OMWU) also enjoy last-iterate convergence (see Lee et al. (2021)), they have not been examined in infinite-horizon games. Therefore, we consider ED as the basic dynamic to construct a new method for solving offline games.

## 3 PRELIMINARIES

### 3.1 PROBLEM FORMULATION

**Deterministic two-player zero-sum Markov games.** An infinite-horizon two-player zero-sum Markov game (Littman, 1994; Shapley, 1953) is represented by a tuple $\mathcal{MG} = (\mathcal{S}, \mathcal{A}, \mathcal{B}, P, r, \gamma)$: $\mathcal{S}$ is the state space. $\mathcal{A}$ is the action space of the max-player, who aims to maximize the cumulative reward. $\mathcal{B}$ is the action space of the min-player, who aims to minimize the cumulative reward. $P \in [0,1]^{|\mathcal{S}||A||B| \times |\mathcal{S}|}$ is the transition probability matrix. $r \in [0,1]^{|\mathcal{S}||A||B|}$ is the reward vector. $\gamma \in (0,1]$ is the discount factor.

In this paper, we examine the deterministic two-player zero-sum MGs with $P \in \{0,1\}^{|\mathcal{S}||A||B| \times |\mathcal{S}|}$, which means that the transition is deterministic. Capable of describing real-world games like chess and Go, it can be viewed as a multi-agent extension to the deterministic MDP (Castro, 2020).

**Policy and value functions.** We use $(\mu, \nu)$ to denote the joint policy, where $\mu$ is the policy of the (first) max-player and $\nu$ is the policy of the (second) min-player. Specifically, $\mu(s) \in \Delta(\mathcal{A})$ ($\nu(s) \in \Delta(\mathcal{B})$) is the max-player's (min-player's) action distribution at state $s \in \mathcal{S}$, with $\mu(s,a)$ ($\nu(s,a)$) being the probability of selecting action $a \in \mathcal{A}$ ($b \in \mathcal{B}$). Furthermore, as in single-agent MDPs, define value functions $V^{\mu,\nu}(s) = \mathbb{E}\left[\sum_{t=0}^{\infty} \gamma^t r(s_t, a_t, b_t) | s_0 = s; \mu, \nu\right]$ and $Q^{\mu,\nu}(s,a,b) = \mathbb{E}\left[\sum_{t=0}^{\infty} \gamma^t r(s_t, a_t, b_t) | s_0 = s, a_0 = a, b_0 = b; \mu, \nu\right]$.

**Nash equilibrium.** A Nash equilibrium (NE) in a game corresponds to a joint policy where each individual player cannot benefit from unilaterally deviating from his/her own policy. Specifically, in a two-player zero-sum MG, an NE $(\mu^*, \nu^*)$ satisfies $V^{\mu,\nu^*} \leq V^{\mu^*,\nu^*} \leq V^{\mu^*,\nu}$ for any $\mu$ and $\nu$. As is well known, every two-player zero-sum MG has at least one NE, and all NEs share the same value:

$$V^*(s) = V^{\mu^*,\nu^*}(s) = \max_{\mu} \min_{\nu} V^{\mu,\nu}(s) = \min_{\nu} \max_{\mu} V^{\mu,\nu}(s)$$

For fixed $\mu$ and $\nu$, define best-response value functions $V^{\mu,*}(s) = \min_{\nu} V^{\mu,\nu}(s)$ and $V^{*,\nu}(s) = \max_{\mu} V^{\mu,\nu}(s)$. Furthermore, let $\rho_0 \in \Delta(\mathcal{S})$ be an initial state distribution and define:

$$\text{NashConv}(\mu, \nu) = \mathbb{E}_{s \sim \rho_0}\left[V^{\mu,\nu}(s) - V^{\mu,*}(s) + V^{*,\nu}(s) - V^{\mu,\nu}(s)\right] = \mathbb{E}_{s \sim \rho_0}\left[V^{*,\nu}(s) - V^{\mu,*}(s)\right]$$

In two-player zero-sum games, NashConv is the sum of the *exploitability* of the players' policies. It also corresponds to the *duality gap* defined from the minimax perspective. For any NE $(\mu^*, \nu^*)$, we have $\text{NashConv}(\mu^*, \nu^*) = 0$. In this paper, we aim to find approximate Nash equilibria, which are joint policies with NashConv close to zero. An important property of NE in two-player zero-sum games is that if $(\mu_1, \nu_1)$ and $(\mu_2, \nu_2)$ are both NEs, then $(\mu_1, \nu_2)$ and $(\mu_2, \nu_1)$ are also NEs. Therefore, it is reasonable to unilaterally learn the equilibrium policy for the max-player and the min-player. Then, an NE can be directly constructed from the individual policies.

### 3.2 EXPLOITABILITY DESCENT

Exploitability Descent (ED) (Lockhart et al., 2019) is a game-theoretic approach that generalizes the classic convex-concave optimization for solving matrix games. The core idea is to iteratively update the current policy along the gradient computed against a best response from the opponent. Compared to the methods of fictitious play (Brown, 1951) and regret minimization (Hart & Mas-Colell, 2000), ED exhibits *last-iterate convergence* rather than *average-iterate convergence* in two-player zero-sum games. Therefore, ED can be readily extended to online deep reinforcement learning algorithms with

---

**Algorithm 1:** Exploitability Descent (ED)

---

**Input:** Game model $\mathcal{MG}$ and iteration number $K$

1 Initialize $\mu_0$

2 **for** $k \in \{1, 2, \cdots, K\}$ **do**

3    Compute $Q_k = Q^{\mu_{k-1}, \nu^\dagger}$ under $\mathcal{MG}$, where $\nu^\dagger = \text{br}(\mu_{k-1})$ is a best response against $\mu_{k-1}$

4    **for** $s \in \mathcal{S}$ **do**

5      Update

$$\mu_k(s) = \underset{\mu(s) \in \Delta(\mathcal{A})}{\arg\min} \left\{ \sum_{a \in \mathcal{A}} \left( \mu(s, a) - \left( \mu_{k-1}(s, a) + \alpha \sum_{b \in \mathcal{B}} \nu^\dagger(s, b) Q_k(s, a, b) \right) \right)^2 \right\}$$

6    **end**

7 **end**

**Output:** Last iterate $\mu_K$ for max-player

---

policies parameterized by neural networks. In two-player zero-sum Markov games, ED for learning max-player's policy $\mu$ is shown in Algorithm 1.

Define the utility function $u(\mu, \nu) = \mathbb{E}_{s_0 \sim \rho_0} [V^{\mu, \nu}(s_0)]$. For each $(s, a)$, $\sum_{b \in \mathcal{B}} \nu^\dagger(s, b) Q_k(s, a, b)$ can make up a generalized gradient of $\mu_{k-1}$'s worst-case utility $\nabla_{\mu(s,a)} u(\mu, \text{br}(\mu)) \in \partial \min_\nu u(\mu, \nu)$ (Clarke, 1975). Following the generalized gradient, $\mu_k$ can approach a local optimum $\hat{\mu}$ of the minimax problem $\max_\mu \min_\nu u(\mu, \nu)$. To optimize min-player's policy, we can exchange $\mu$ and $\nu$ in Algorithm 1 and use $-\alpha$ on line 5. Then, $(\hat{\mu}, \hat{\nu})$ constructs a potential Nash equilibrium.

### 3.3 POLICY CONSTRAINT METHODS

In offline RL, the training process is always affected by action distributional shift (Kumar et al., 2019), which is one of the largest obstacles for model-free application of learning dynamics like Algorithm 1. In single-agent scenarios, the effect can be weakened by applying constraints to the learned policy $\pi$ to keep it close to the behavior policy $\pi_\beta$, which follows the distribution of the offline data. This ensures that the process of Q-function computation hardly queries the out-of-distribution actions. The accumulative error in value estimation can be avoided at the expense of policy suboptimality.

Such constraints are commonly realized using direct *policy constraints* on the policy update (Siegel et al., 2020) or indirect *policy penalties* on the value functions (Wu et al., 2019). Both methods require using certain measure $D(\cdot, \cdot)$ (e.g., KL-divergence) to describe the closeness of two policies.

The following policy update formula is an example of applying direct policy constraints:

$$\pi_k(s) = \underset{\pi(s)}{\arg\max} \left\{ \mathbb{E}_{a \sim \pi(s)} [Q^{\pi_k}(s, a)] \right\} \quad \text{s.t. } D(\pi(s), \pi_\beta(s)) \leq \delta$$

In comparison, a regularized value is computed when using indirect policy penalties:

$$\pi_k(s) = \underset{\pi(s)}{\arg\max} \left\{ \mathbb{E}_{a \sim \pi(s)} [Q^{\pi_k}(s, a)] - \epsilon D(\pi(s), \pi_\beta(s)) \right\}$$

For direct policy constraints, the optimality of the learned policy is preserved only when the behavior policy $\pi_\beta$ is close enough to the true optimal policy, which is in theory a pure strategy in single-agent scenarios. However, this is unlikely to happen since $\pi_\beta$ is derived from an offline dataset. For indirect policy penalties, we will see that they face the same problem since the ultimate solution could never be a pure strategy (see Lemma 1 for the case of KL-divergence).

## 4 CONSTRAINED EXPLOITABILITY DESCENT

For adversarial games, even if we only apply a constraint to the computation of the best response $\nu^\dagger$ for the min-player in Algorithm 1, the resulting max-player policy $\mu$ will surely deviate from the equilibrium of the original game for the same reason in single-agent scenarios. However, we find

that it is possible to instead keep the min-player policy $\nu$ unexploitable. We will further explain it through our subsequent mathematical derivations in Section 5. With this observation, we propose an offline equilibrium-learning algorithm named Constrained Exploitability Descent (CED) under policy constraints (Algorithm 2).

---

**Algorithm 2:** Constrained Exploitability Descent (CED)

---

**Input:** Offline dataset $\mathcal{D}$, discount factor $\gamma$, and iteration number $K$

1  Set policy constraint measure $D(\cdot, \cdot)$ and range $\delta$, policy penalty parameter $\epsilon$, and learning rate $\alpha$
2  Extract non-repetitive transition set $\mathcal{D}^*$, state set $\mathcal{S}$, and action sets $\mathcal{A}, \mathcal{B}$ from $\mathcal{D}$
3  Compute state distribution $\rho_\mathcal{D}$ and behavior policy $(\mu_\beta, \nu_\beta)$ from $\mathcal{D}$
   `// Evaluate the value function under behavior policy`
4  Compute $Q^{\mu_\beta, \nu_\beta} =$

$$\arg\min_Q \left\{ \sum_{(s,a,b,r,s') \in \mathcal{D}^*} \left( Q(s,a,b) - \left( r(s,a,b) + \gamma \mathbb{E}_{\substack{a' \sim \mu_\beta(s') \\ b' \sim \nu_\beta(s')}} \left[ Q(s',a',b') \right] \right) \right)^2 \right\}$$

5  Initialize $Q_0 = Q^{\mu_\beta, \nu_\beta}, \mu_0 = \mu_\beta, \nu_0 = \nu_\beta$
6  **for** $k \in \{1, 2, \cdots, K\}$ **do**
      `// Apply Bellman operator to the current value function`
7  │  Update $Q_k =$

$$\arg\min_Q \left\{ \sum_{(s,a,b,r,s') \in \mathcal{D}^*} \left( Q(s,a,b) - \left( r(s,a,b) + \gamma \mathbb{E}_{\substack{a' \sim \mu_{k-1}(s') \\ b' \sim \nu_{k-1}(s')}} \left[ Q_{k-1}(s',a',b') \right] \right) \right)^2 \right\}$$

   `// Update `$\mu$` along ED-like gradient under policy constraint`
8  │  **for** $s \in \mathcal{S}$ **do**
9  │  │  Under constraint $D(\mu(s), \mu_\beta(s)) \leq \delta$, update $\mu_k(s) =$

$$\arg\min_{\mu(s) \in \Delta(\mathcal{A})} \left\{ \sum_{a \in \mathcal{A}} \left( \mu(s,a) - \left( \mu_{k-1}(s,a) + \alpha \rho_\mathcal{D}(s) \sum_{b \in \mathcal{B}} \nu_{k-1}(s,b) Q_k(s,a,b) \right) \right)^2 \right\}$$

10 │  **end**
   `// Compute approximate best response `$\nu$` under policy penalty`
11 │  **for** $s \in \mathcal{S}$ **do**
12 │  │  Compute $\nu_k(s) =$

$$\arg\max_{\nu(s) \in \Delta(\mathcal{B})} \left\{ \sum_{b \in \mathcal{B}} \nu(s,b) \left( - \sum_{a \in \mathcal{A}} \mu_k(s,a) Q^{\mu_\beta, \nu_\beta}(s,a,b) \right) - \epsilon D_{\mathrm{KL}} \left( \nu(s), \nu_\beta(s) \right) \right\}$$

13 │  **end**
14 **end**
   **Output:** Last iterate $\nu_K$ for min-player

---

CED inherits the basic structure of ED in each iteration. A $Q$ value is computed, the current $\mu$ is updated, and a best response $\nu$ is computed in preparation for the next iteration. However, CED has multiple differences in detail:

- $Q_k$ is based on the last $Q_{k-1}$ rather than directly solved under the current Bellman equation.

- The update of $\mu$ at each state $s \in \mathcal{S}$ is under a direct policy constraint $D(\mu(s), \mu_\beta(s)) \leq \delta$. An additional factor $\rho_\mathcal{D}(s)$ is also appended after the learning rate $\alpha$.

- The computation of $\nu$ is based on $Q^{\mu_\beta, \nu_\beta}$ (without estimating $Q^{\mu_k, \nu_k}$) and under a KL-divergence penalty $D_{\mathrm{KL}}(\nu(s), \nu_\beta(s))$ with a regularization parameter $\epsilon$.

Note that $\nu_k$ can still be viewed as an approximate best response to the current $\mu_k$ when $(\mu_k, \nu_k)$ is kept close to $(\mu_\beta, \nu_\beta)$. As a result, the last iterate $\mu_K$ locally minimizes exploitability in a regularized game. However, under the additional KL-divergence regularization, now $\nu_k$ has a unique solution with a closed-form expression (see Lemma 1), which allows $\mu$ to update along a deterministic gradient rather than an arbitrary generalized gradient. This mitigates the problem that following a generalized gradient can lead to recurrence around a local optimum (see the experimental result in Section 6.2).

**Lemma 1** (Uniqueness of $\nu$ in CED).

$$\nu_k(s,b) = \frac{\nu_\beta(s,b) \exp\left(-\frac{1}{\epsilon}\sum_{a\in\mathcal{A}}\mu_k(s,a)Q^{\mu_\beta,\nu_\beta}(s,a,b)\right)}{\sum_{b'\in\mathcal{B}}\nu_\beta(s,b')\exp\left(-\frac{1}{\epsilon}\sum_{a\in\mathcal{A}}\mu_k(s,a)Q^{\mu_\beta,\nu_\beta}(s,a,b')\right)}$$

When $\epsilon > 0$, the min-player policy $\nu$ is a mixed strategy and no longer an exact best response to the max-player policy $\mu$. As a result, the limit point of $\mu$ deviates from the solution to the original minimax problem. Instead, we will prove in the following section that $\nu_k$ approaches an unexploitable $\hat{\nu}$. By exchanging the status of max-player and min-player in the game and running Algorithm 1 again, we can also obtain an unexploitable $\hat{\mu}$ with an independent run of CED. The joint policy $(\hat{\mu}, \hat{\nu})$ will construct a potential Nash equilibrium.

## 5 THEORETICAL ANALYSIS

In this section, we theoretically show that it is possible for CED (Algorithm 2) to find an exact Nash equilibrium with the following two steps: First, we prove that CED can converge to a stationary point $(\bar{Q}, \bar{\mu}, \bar{\nu})$ (Section 5.1). Second, we prove that the min-player policy $\bar{\nu}$ at the stationary point of CED is unexploitable, like any mixed-strategy Nash equilibrium of full support (Section 5.2). All of the omitted proofs are provided in Appendix A.

Throughout our analysis, we require the *uniform coverage* assumption, which means that the non-repetitive transition set $\mathcal{D}^*$ derived from the dataset $\mathcal{D}$ covers all state-action tuples $(s, a, b)$. In Cui & Du (2022a), this assumption is called uniform concentration, and a weaker assumption named unilateral concentration is analyzed. By constructing a counterexample where the exact NE becomes impossible to learn, they proved that unilateral concentration is somewhat necessary for finding Nash equilibrium offline. However, when the NE is a completely mixed strategy (e.g., the unique NEs of the matrix games in Section 6.1), unilateral concentration is equivalent to uniform concentration. Therefore, the uniform coverage assumption can be necessary for our theoretical analysis on finding mixed-strategy Nash equilibrium.

### 5.1 CONVERGENCE OF CED

Lemma 2 gives the explicit expression on the gradient of utility function $u(\mu, \nu) = \mathbb{E}_{s\sim\rho_0}\left[V^{\mu,\nu}(s)\right]$ with respect to $\mu$. This can be viewed as an application of the policy gradient theorem in MDPs (Sutton et al., 1999) to multi-agent scenarios.

**Lemma 2** (Policy Gradient in MG). *Let* $\rho^{\mu,\nu}(s) = \sum_{k\geq 0}\gamma^k\Pr(s|k;\mu,\nu)$, *where* $\Pr(s|k;\mu,\nu)$ *is the probability of reaching state $s$ at time step $k$ under joint policy $(\mu, \nu)$. Then, it holds:*

$$\frac{\partial u(\mu,\nu)}{\partial\mu(s,a)} = \rho^{\mu,\nu}(s)\sum_{b\in\mathcal{B}}\nu(s,b)Q^{\mu,\nu}(s,a,b) \quad (\forall s\in\mathcal{S}, a\in\mathcal{A})$$

Using Lemma 1 and Lemma 2, we are able to demonstrate the convergence of CED (Theorem 1) under an approximation about the state visitation probability $\rho$.

**Theorem 1** (Convergence of CED). *When $\rho^{\mu,\nu}$ approximates the true state distribution $\rho_\mathcal{D}$ of the dataset $\mathcal{D}$, CED with sufficiently small $\alpha$ and $\frac{1}{\epsilon}$ will converge to a stationary point $(\bar{Q}, \bar{\mu}, \bar{\nu})$ under uniform coverage assumption.*

*Proof.* By Lemma 1, $\nu_k$ is uniquely determined by $\mu_k$. As $\mathcal{D}^*$ covers all $(s, a, b)$ tuples and the MG is deterministic, $Q_{k+1}$ in CED approximates the true value $Q^{\mu_k,\nu_k}$ when $\mu$'s learning rate $\alpha$ is close to zero. Therefore, we only need to consider the convergence of $\mu$. By Lemma 2, we have:

$$\frac{\partial u(\mu_k,\nu(\mu_k))}{\partial\mu_k(s,a)} = \frac{\partial u(\mu_k,\nu_k)}{\partial\mu_k(s,a)} + \sum_{b\in\mathcal{B}}\frac{\partial u(\mu_k,\nu_k)}{\partial\nu_k(s,b)}\frac{\partial\nu_k(s,b)}{\partial\mu_k(s,a)} =$$

$$\sum_{b\in\mathcal{B}}\left(\rho^{\mu_k,\nu_k}(s)\nu_k(s,b)Q^{\mu_k,\nu_k}(s,a,b) + \frac{\partial u(\mu_k,\nu_k)}{\partial\nu_k(s,b)}\frac{\partial\nu_k(s,b)}{\partial\mu_k(s,a)}\right)$$

Note that $\frac{\partial \nu_k(s,b)}{\partial \mu_k(s,a)} \to 0$ when $\frac{1}{\epsilon} \to 0$ (see Appendix A.3 for details). When $\rho^{\mu,\nu}$ approximates $\rho_{\mathcal{D}}$, we have $\frac{\partial u(\mu_k,\nu_k)}{\partial \mu_k(s,a)} = \rho_{\mathcal{D}}(s) \sum_{b \in \mathcal{B}} \nu_k(s,b) Q_{k+1}(s,a,b)$. Therefore, $\mu_k$ in CED updates along the gradient of $u(\mu, \nu(\mu))$ at a sufficiently small learning rate $\alpha$. As a result, $\mu$ will converge to a local maximum $\bar{\mu}$ for $u(\mu, \nu(\mu))$, which implies CED will converge to a stationary point $(\bar{Q}, \bar{\mu}, \bar{\nu})$. □

Theorem 1 provides a direct convergence guarantee for CED without relying on a generalized gradient like ED. Besides, compared to ED's underlying assumption that $\rho^{\mu,\nu}$ is uniform, the assumption of $\rho^{\mu,\nu} \approx \rho_{\mathcal{D}}$ is more realistic. The policy constraints employed in CED will keep $(\mu_k, \nu_k)$ close to the behavior policy $(\mu_\beta, \nu_\beta)$ derived from $\mathcal{D}$. Thus, the visitation probabilities can be close as well.

## 5.2 Relationship to Nash Equilibrium

Now we further show that the min-player policy $\bar{\nu}$ at the stationary point of CED satisfies an inherent property of the mixed-strategy Nash equilibria, namely, being unexploitable.

**Definition 1** (Unexploitable). *We say a joint policy $(\mu, \nu)$ in an MG is unexploitable if $\mu$ and $\nu$ are both unexploitable with respect to each other. Specifically, $\forall s \in \mathcal{S}$:*

$$\sum_{a \in \mathcal{A}} \mu(s,a) Q^{\mu,\nu}(s,a,b) = c_s, \forall b \in \mathcal{B} \text{ means that } \mu \text{ is unexploitable with respect to } \nu.$$

$$\sum_{b \in \mathcal{B}} \nu(s,b) Q^{\mu,\nu}(s,a,b) = c_s, \forall a \in \mathcal{A} \text{ means that } \nu \text{ is unexploitable with respect to } \mu.$$

Intuitively, a policy $\mu$ is unexploitable with respect to an opponent policy $\nu$ in an MG if the opponent has the same value $c_s$ for all actions under each $s \in \mathcal{S}$. As a result, the opponent cannot exploit $\mu$ by deviating from $\nu$ at any state. We use Lemma 3 to show that this property can characterize the mixed-strategy Nash equilibria with full support.

**Lemma 3** (Property of Interior NE). *If a Nash equilibrium $(\mu^*, \nu^*)$ in an MG has full support on the action space (thus being an interior point of the joint policy space), then $(\mu^*, \nu^*)$ is unexploitable.*

Now we start to demonstrate that $\bar{\nu}$ at any stationary point of CED is also an unexploitable min-player policy in the MG. We first provide an auxiliary lemma that shows the update of $\mu$ at each state $s \in \mathcal{S}$ can be equivalently enforced within the hyperplane of the probability simplex, where $\sum_{a \in A} \mu(s,a) = 1$.

**Lemma 4** (Update Projection). *Let $z_a^s = \alpha \rho_{\mathcal{D}}(s) \sum_{b \in \mathcal{B}} \nu_k(s,b) Q_{k+1}(s,a,b)$ be the original update for $\mu_k(s,a)$ in CED. Let $y = \sum_{a \in \mathcal{A}} z_a^s$ be the summation over $\mathcal{A}$ and define the projected update as $p_a^s = z_a^s - \frac{y}{|\mathcal{A}|}$. Then, replacing all $z_a^s$ with $p_a^s$ results in the same $\mu_{k+1}(s)$ in CED.*

We call $p_a^s$ projected update since $\sum_{a \in \mathcal{A}} p_a^s = 0$ and $(\mu(s,a) + p_a^s)_{a \in \mathcal{A}}$ is kept in the hyperplane of the probability simplex. Using Lemma 4, we can prove that $\bar{\nu}$ is unexploitable under an interior point assumption, which is also sufficient for the theoretical analysis of ED (Lockhart et al., 2019).

**Theorem 2** (Unilateral Unexploitability). *Let $\Pi(s) = \Pi_1(s) \cap \Pi_2(s)$ be the feasible region for $\mu(s)$, where $\Pi_1(s)$ is the probability simplex and $\Pi_2(s)$ is the region induced by the constraint $D(\mu(s), \mu_\beta(s)) \leq \delta$. For any stationary point $(\bar{Q}, \bar{\mu}, \bar{\nu})$ of CED, if $\bar{\mu}(s)$ is an interior point of $\Pi(s)$ for all $s \in \mathcal{S}$, then $\bar{\nu}$ is an unexploitable policy with respect to $\bar{\mu}$ under uniform coverage assumption.*

*Proof.* As $\mathcal{D}^*$ covers all $(s,a,b)$ tuples and the MG is deterministic, a stable $\bar{Q}$ with respect to $(\bar{\mu}, \bar{\nu})$ in CED corresponds to the true value $Q^{\bar{\mu},\bar{\nu}}$. Since $(\bar{\mu}(s,a) + p_a^s)_{a \in \mathcal{A}}$ is in the hyperplane of $\Pi(s)$ and $\bar{\mu}$ is stable with respect to $(\bar{Q}, \bar{\nu})$, we can consider the following two cases:

- $(\bar{\mu}(s,a) + p_a^s)_{a \in \mathcal{A}}$ belongs to $\Pi(s)$. Then, $\bar{\mu}(s) = (\bar{\mu}(s,a) + p_a^s)_{a \in \mathcal{A}} \Rightarrow p_a^s = 0, \forall a \in \mathcal{A}$.

- $(\bar{\mu}(s,a) + p_a^s)_{a \in \mathcal{A}}$ does not belong to $\Pi(s)$. Then, $\bar{\mu}(s)$ is the closest point in $\Pi(s)$ with respect to the point $(\bar{\mu}(s,a) + p_a^s)_{a \in \mathcal{A}}$ in the same hyperplane. This contradicts the assumption that $\bar{\mu}(s)$ is an interior point of $\Pi(s)$.

Therefore, it holds for all $s \in \mathcal{S}$ that $p_a^s = 0, \forall a \in \mathcal{A}$, which further implies that $z_a^s = c_s, \forall a \in \mathcal{A}$. As a result, $\sum_{b \in \mathcal{B}} \bar{\nu}(s, b)\bar{Q}(s, a, b) = \sum_{b \in \mathcal{B}} \bar{\nu}(s, b)Q^{\bar{\mu},\bar{\nu}}(s, a, b) = c_s, \forall a \in \mathcal{A}$, which means that the min-player policy $\bar{\nu}$ is unexploitable with respect to $\bar{\mu}$. $\qquad\square$

With Theorem 2, if we run Algorithm 2 twice by exchanging the status of the two players and both max-player policies converge to an interior point, then the last iterates $(\mu, \hat{\nu})$ and $(\hat{\mu}, \nu)$ can construct an unexploitable joint policy $(\hat{\mu}, \hat{\nu})$. Policy constraints play an important role in supporting this claim. On the one hand, the distance between $\mu$ and $\mu_\beta$ is restricted by the direct policy constraint. On the other hand, the indirect policy penalty can also bound the distance between $\hat{\mu}$ and $\mu_\beta$ (corresponding to the $\nu_k$ and $\nu_\beta$ in Algorithm 2 after the status exchange; see Lemma 5 in Appendix A.6 for an explicit bound). Since both $\mu$ and $\hat{\mu}$ are close to $\mu_\beta$ under policy constraints, we have $Q^{\mu,\hat{\nu}} \approx Q^{\mu_\beta,\hat{\nu}} \approx Q^{\hat{\mu},\hat{\nu}}$, which implies that $\hat{\nu}$ is also unexploitable with respect to $\hat{\mu}$. By symmetry, it is direct to show that the joint policy $(\hat{\mu}, \hat{\nu})$ is unexploitable and thus constructs a potential mixed-strategy Nash equilibrium.

In Appendix C.1, we combine the existing theory to provide an overall explanation on the CED method. In the next section, we will further verify through experiments that CED can practically find NE policies under uniform coverage. Even if the data coverage is non-uniform, we still find that CED can gradually improve the behavior policy from the offline dataset.

## 6 EXPERIMENTS

Here we conduct experiments for CED in matrix games, a tree-form game, and a soccer game. Each single run of CED can be finished within one hour using a single Intel Core i7-12700F CPU.

### 6.1 MATRIX GAME

We first examine if CED manages to find mixed-strategy Nash equilibrium in static matrix games. We consider two games with two valid actions from $\{1, 2\}$ for both players. The payoff matrices are $\mathcal{M}_1 = \begin{pmatrix} 1 & 0 \\ -2 & 4 \end{pmatrix}$ and $\mathcal{M}_2 = \begin{pmatrix} 1 & 0 \\ -2 & 3 \end{pmatrix}$, respectively, where the rows correspond to the actions of the max-player and the columns correspond to the actions of the min-player. The unique NE of $\mathcal{M}_1$ is $\left(\mu^*(1) = \frac{6}{7}, \nu^*(1) = \frac{4}{7}\right)$, and the unique NE of $\mathcal{M}_2$ is $\left(\mu^*(1) = \frac{5}{6}, \nu^*(1) = \frac{1}{2}\right)$.

The learning curves of $(\mu, \nu)$ in a single execution of CED ($\alpha = 0.01, \epsilon = 1.0$) under uniform coverage $\left(\mu_\beta(1) = \frac{1}{2}, \nu_\beta(1) = \frac{1}{2}\right)$ are shown in Figure 1. The y-axis indicates the probability of choosing action 1 under the corresponding policy. The dashed line indicate the unique NE policy. In both games, CED manages to learn the equilibrium policy $\nu = \nu^*$ for the min-player. This result is consistent with Theorem 1 and Theorem 2, which claim that under uniform coverage, CED will converge to an unexploitable $\nu$ (an NE policy in this case). We may find that the learned $\mu$ for $\mathcal{M}_2$ also corresponds to the equilibrium. However, this is because $\nu_\beta$ happens to be $\nu^*$ in $\mathcal{M}_2$. Otherwise, the divergence regularization applied to the computation of $\nu$ will force the stationary point of $\mu$ to deviate from $\mu^*$ because $\nu^*$ is not an exact best response to the convergent $\mu$. This phenomenon is shown in the learning curve on $\mathcal{M}_1$, with the ultimate $\mu \neq \mu^*$ as a result of $\nu_\beta \neq \nu^*$.

We also test CED in a 5-action "Rock-Paper-Scissors-Fire-Water" game denoted by $\mathcal{M}_3$. Besides the common rules of the RPS game, fire beats everything except water, and water is beaten by everything except it beats fire. $\left(\frac{1}{9}, \frac{1}{9}, \frac{1}{9}, \frac{1}{3}, \frac{1}{3}\right)$ is an unexploitable policy for both players, and the unique Nash equilibrium of $\mathcal{M}_3$ is constructed when both use this policy. As is shown in Figure 1 (right), CED ($\alpha = 0.01, \epsilon = 0.1$) manages to learn the mixed-strategy equilibrium policy.

### 6.2 TREE-FORM GAME

Now we further consider dynamic games, where the Nash equilibrium at a decision point is affected by the results of subsequent game stages. We examine the learning behaviors in a tree-form game $\mathcal{T}$ consisting of three decision points whose payoff matrices are $\mathcal{M}_1$, $\mathcal{M}_2$, and $\mathcal{M}_3$, respectively. $\mathcal{T}$ starts with Stage 1 ($\mathcal{M}_1$) and enters Stage 2 ($\mathcal{M}_2$) or Stage 3 ($\mathcal{M}_3$) conditioned on the joint actions of two players at Stage 1 (see Appendix B.1). By backward induction, we can compute that the NE at Stage 1 is $\left(\mu^*(1) = \frac{13}{16}, \nu^*(1) = \frac{9}{16}\right)$, which deviates from the original equilibrium of $\mathcal{M}_1$.

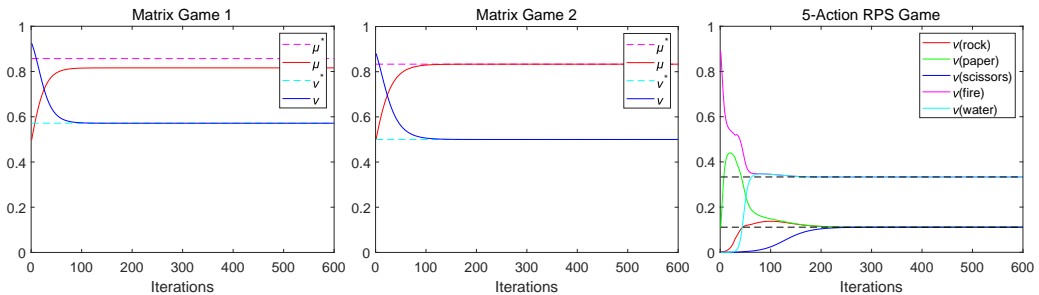

Figure 1: CED learning curves in matrix games

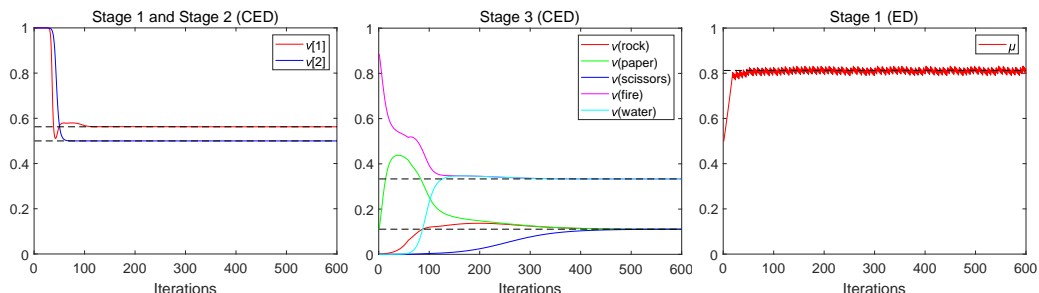

Figure 2: CED / ED learning curves in the tree-form game

As is shown in Figure 2 (left & mid), CED ($\alpha = 0.005, \epsilon = 0.1$) finds the NE policy for the min-player in the tree-form game. As there is a mismatch between the convergence speed at Stage 2 and Stage 3, $\nu$ at Stage 1 experiences an oscillation and eventually converges to the solution. This phenomenon is consistent with the intuition that the learning process at the initial stage depends on subsequent stages in dynamic games. Besides, we test the behavior of the model-based ED algorithm in this scenario. As is shown in Figure 2 (right), while ED can approximate the NE policy for the max-player, it suffers from continual oscillations as a side effect of following generalized gradient.

## 6.3 SOCCER GAME

While the theoretical analysis and the toy problem experiments above have suggested the capability of CED to find mixed-strategy Nash equilibrium, here we further verify the conclusion in an infinite-horizon Markov game, i.e., the soccer game (see Appendix B.2). To measure the performance of CED, we compute the NashConv of the learned $(\mu, \nu)$ and compare it with the result of a pessimistic model-based algorithm, VI-LCB-Game (Yan et al., 2024), which provably finds approximate Nash equilibrium offline for infinite-horizon MGs but requires infinitely many samples in theory. In Figure 3 (left), the dashed line shows the NashConv of the joint policy derived from VI-LCB-Game, given the minimum amount of samples for uniform coverage. Under the same offline dataset, CED ($\alpha = 10^{-6}, \epsilon = 10^{-3}$) steadily reduces the exploitability of the learned policy and eventually obtains a policy with significantly lower NashConv.

In Theorem 1, the convergence of CED theoretically relies on sufficiently small $\alpha$ and $\frac{1}{\epsilon}$. Thus, we also examine the practical behavior of CED under different $\alpha$ and $\epsilon$. As is shown in Figure 3 (mid), an overly large $\alpha$ makes it significantly harder for CED to converge, while an overly small $\alpha$ slows down the speed of learning. Figure 3 (right) also shows that the regularization parameter $\epsilon$ should not be too small. These results match our theoretical analysis and suggest that the conditions on $\alpha$ and $\epsilon$ in Theorem 1 could be necessary as well.

As CED is model-free and does not rely on the full game information, it is in principle applicable to an arbitrary set of offline data, regardless of the coverage. Here we further examine if it can gradually improve the behavior policy when the coverage is non-uniform, like those single-agent offline RL algorithms. To be specific, we randomly banned one action out of five for each player at each state and removed all the related transitions from the dataset $\mathcal{D}$. This makes it impossible to learn an exact

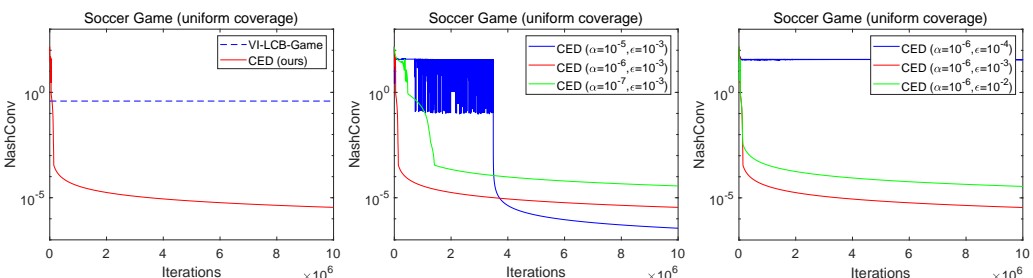

Figure 3: Learning curve comparison in the soccer game under uniform coverage

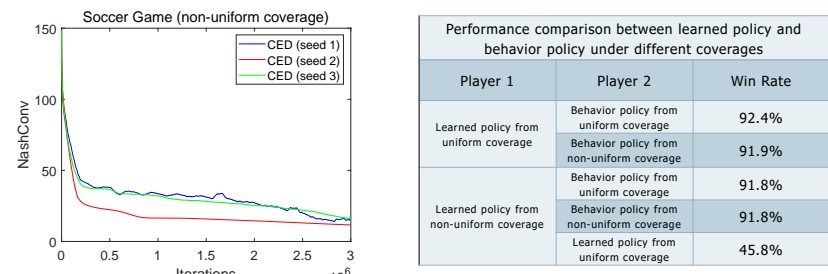

Figure 4: Performance improvement over behavior policy by CED in the soccer game

Nash equilibrium in theory, as a preferred action from the NE can be completely removed. Still, CED gradually improves the behavior policy under such data coverage, as is shown in Figure 4 (left).

Besides NashConv, we estimate the win rate to intuitively show the improvement of learned policy over behavior policy by CED. As is shown in Figure 4 (right), whether under uniform or non-uniform coverage, the policy learned by CED significantly improves the practical performance, with win rates over 90% against behavior policies. It is a little surprising that while the NashConv achieved by CED from non-uniform coverage is much higher than that from uniform coverage, the gap is not that much with respect to the win rate. This reflects that CED can still learn a practically competitive policy even from offline datasets without uniform coverage. Appendix C.2 provides a further discussion on the performance of CED under non-uniform coverage.

# 7 CONCLUSION

In this paper, by proposing CED and analyzing its convergence properties, we demonstrate for the first time that, unlike in MDPs, an optimal policy can be learned under policy constraints in adversarial MGs. This conclusion is drawn from our theoretical and empirical results. With Theorem 1 and Theorem 2, we prove that under uniform coverage, CED converges to an unexploitable min-player policy without relying on the generalized gradient. In the experiments, our theory is verified by the practical results of CED in multiple game scenarios. We also show that, similar to single-agent offline RL algorithms, CED can improve the behavior policy even from datasets without uniform coverage.

We hope this work will inspire more research on solving offline games. Actually, since CED is constructed based on the game-theoretic approach of exploitability descent, which is also capable of solving imperfect-information games (IIGs), it is possible to use CED as an offline IIG solver by replacing the state and value with information state and counterfactual value. However, how to estimate counterfactual value based on the current policy and offline game data remains an open problem. In order to guarantee a stable performance, further theoretical analysis is still required.

CED has the limitation that it is only able to find the mixed-strategy Nash equilibria in two-player zero-sum games. However, it may not be the unique way of learning Nash equilibrium under policy constraints, as a wide range of algorithms that exhibit last-iterate convergence (e.g., OMWU (Lee et al., 2021)) are currently available in the field of game theory. Combining them with existing offline

RL techniques may lead to more offline RL algorithms with possibly better convergence guarantees and practical equilibrium-finding capabilities.

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

# A OMITTED PROOFS

## A.1 PROOF OF LEMMA 1

*Proof.* First, we prove:

$$\pi = \underset{\pi \in \Delta(\mathcal{A})}{\arg\max} \left\{ \sum_{a \in \mathcal{A}} \pi(a) \left( r(a) - \log \pi(a) \right) \right\} \Rightarrow \pi(a) \propto e^{r(a)}$$

Write the corresponding optimization problem:

$$\begin{cases} \text{maximize } \sum_{a \in \mathcal{A}} \pi(a) \left( r(a) - \log \pi(a) \right) \\ \text{s.t.} \quad \sum_{a \in \mathcal{A}} \pi(a) = 1 \\ \quad \pi(a) \geq 0, \ \forall a \in \mathcal{A} \end{cases}$$

Using the Lagrange multiplier, we have:

$$L = \sum_{a \in \mathcal{A}} \pi(a) \left( r(a) - \log \pi(a) \right) - \lambda \left( \sum_{a \in \mathcal{A}} \pi(a) - 1 \right)$$

$$\frac{\partial L}{\partial \pi(a)} = 0 \Rightarrow r(a) - \left( \log \pi(a) + \frac{\pi(a)}{\pi(a)} \right) - \lambda = 0$$

$$\Rightarrow \pi(a) = e^{r(a) - \lambda - 1} \Rightarrow \pi(a) \propto e^{r(a)}$$

By definition of $\nu_k$, we have:

$$\nu_k(s) = \underset{\nu(s) \in \Delta(\mathcal{B})}{\arg\max} \left\{ \sum_{b \in \mathcal{B}} \nu(s,b) \left( - \sum_{a \in \mathcal{A}} \mu_k(s,a) Q^{\mu_\beta, \nu_\beta}(s,a,b) \right) - \epsilon D_{\mathrm{KL}} \left( \nu(s), \nu_\beta(s) \right) \right\}$$

$$= \underset{\nu(s) \in \Delta(\mathcal{B})}{\arg\max} \left\{ \sum_{b \in \mathcal{B}} \nu(s,b) \left( -\frac{1}{\epsilon} \sum_{a \in \mathcal{A}} \mu_k(s,a) Q^{\mu_\beta, \nu_\beta}(s,a,b) - \log \frac{\nu(s,b)}{\nu_\beta(s,b)} \right) \right\}$$

$$= \underset{\nu(s) \in \Delta(\mathcal{B})}{\arg\max} \left\{ \sum_{b \in \mathcal{B}} \nu(s,b) \left( \log \nu_\beta(s,b) - \frac{1}{\epsilon} \sum_{a \in \mathcal{A}} \mu_k(s,a) Q^{\mu_\beta, \nu_\beta}(s,a,b) - \log \nu(s,b) \right) \right\}$$

Therefore:

$$\nu_k(s,b) \propto \exp \left( \log \nu_\beta(s,b) - \frac{1}{\epsilon} \sum_{a \in \mathcal{A}} \mu_k(s,a) Q^{\mu_\beta, \nu_\beta}(s,a,b) \right)$$

which implies:

$$\nu_k(s,b) = \frac{\nu_\beta(s,b) \exp \left( -\frac{1}{\epsilon} \sum_{a \in \mathcal{A}} \mu_k(s,a) Q^{\mu_\beta, \nu_\beta}(s,a,b) \right)}{\sum_{b' \in \mathcal{B}} \nu_\beta(s,b') \exp \left( -\frac{1}{\epsilon} \sum_{a \in \mathcal{A}} \mu_k(s,a) Q^{\mu_\beta, \nu_\beta}(s,a,b') \right)}$$

$\square$

## A.2   PROOF OF LEMMA 2

*Proof.* By definition:

$$\frac{\partial V^{\mu,\nu}(s)}{\partial \mu(\hat{s}, a)} = \frac{\partial}{\partial \mu(\hat{s}, a)} \sum_{a \in \mathcal{A}} \mu(s, a) \sum_{b \in \mathcal{B}} \nu(s, b) Q^{\mu,\nu}(s, a, b)$$

$$= \sum_{a \in \mathcal{A}} \left( \frac{\partial \mu(s, a)}{\partial \mu(\hat{s}, a)} \sum_{b \in \mathcal{B}} \nu(s, b) Q^{\mu,\nu}(s, a, b) + \mu(s, a) \sum_{b \in \mathcal{B}} \nu(s, b) \frac{\partial Q^{\mu,\nu}(s, a, b)}{\partial \mu(\hat{s}, a)} \right)$$

$$= \mathbb{I}[s = \hat{s}] \sum_{b \in \mathcal{B}} \nu(s, b) Q^{\mu,\nu}(s, a, b) + \mu(s, a) \sum_{b \in \mathcal{B}} \nu(s, b) \frac{\partial}{\partial \mu(\hat{s}, a)} \left( r(s, a, b) + \gamma V^{\mu,\nu}(s') \right)$$

$$= \mathbb{I}[s = \hat{s}] \sum_{b \in \mathcal{B}} \nu(s, b) Q^{\mu,\nu}(s, a, b) + \sum_{a \in \mathcal{A}} \mu(s, a) \sum_{b \in \mathcal{B}} \nu(s, b) \gamma \frac{\partial V^{\mu,\nu}(s')}{\partial \mu(\hat{s}, a)}$$

$$= \cdots\cdots$$

$$= \sum_{k=0}^{\infty} \gamma^k \Pr(s \to \hat{s} | k; \mu, \nu) \sum_{b \in \mathcal{B}} \nu(\hat{s}, b) Q^{\mu,\nu}(\hat{s}, a, b)$$

where $\mathbb{I}[\cdot]$ is the indicator function and $\Pr(s \to \hat{s} | k; \mu, \nu)$ is the probability of reaching $\hat{s}$ from $s$ using $k$ steps under joint policy $(\mu, \nu)$.

Then, it is direct to show:

$$\frac{\partial u(\mu, \nu)}{\partial \mu(s, a)} = \frac{\partial}{\partial \mu(s, a)} \mathbb{E}_{s_0 \sim \rho_0} \left[ V^{\mu,\nu}(s_0) \right]$$

$$= \sum_{s_0 \in \mathcal{S}} \rho_0(s_0) \sum_{k=0}^{\infty} \gamma^k \Pr(s_0 \to s | k; \mu, \nu) \sum_{b \in \mathcal{B}} \nu(s, b) Q^{\mu,\nu}(s, a, b)$$

$$= \sum_{k=0}^{\infty} \gamma^k \Pr(s | k; \mu, \nu) \sum_{b \in \mathcal{B}} \nu(s, b) Q^{\mu,\nu}(s, a, b)$$

$$= \rho^{\mu,\nu}(s) \sum_{b \in \mathcal{B}} \nu(s, b) Q^{\mu,\nu}(s, a, b)$$

$\square$

## A.3 Detail in Theorem 1

Here, we will show that $\frac{\partial \nu_k(s,b)}{\partial \mu_k(s,a)} \to 0$ when $\frac{1}{\epsilon} \to 0$.

By Lemma 1:

$$\nu_k(s,b) = \frac{\nu_\beta(s,b) \exp\left(-\frac{1}{\epsilon} \sum_{a \in \mathcal{A}} \mu_k(s,a) Q^{\mu_\beta, \nu_\beta}(s,a,b)\right)}{\sum_{b' \in \mathcal{B}} \nu_\beta(s,b') \exp\left(-\frac{1}{\epsilon} \sum_{a \in \mathcal{A}} \mu_k(s,a) Q^{\mu_\beta, \nu_\beta}(s,a,b')\right)}$$

Besides:

$$\frac{\partial \exp\left(-\frac{1}{\epsilon} \sum_{a \in \mathcal{A}} \mu_k(s,a) Q^{\mu_\beta, \nu_\beta}(s,a,b)\right)}{\partial \mu_k(s,a)} =$$

$$-\frac{1}{\epsilon} Q^{\mu_\beta, \nu_\beta}(s,a,b) \exp\left(-\frac{1}{\epsilon} \sum_{a \in \mathcal{A}} \mu_k(s,a) Q^{\mu_\beta, \nu_\beta}(s,a,b)\right)$$

Therefore:

$$\frac{\partial \nu_k(s,b)}{\partial \mu_k(s,a)} = \frac{1}{\epsilon} \nu_\beta(s,b) \exp\left(-\frac{1}{\epsilon} \sum_{a \in \mathcal{A}} \mu_k(s,a) Q^{\mu_\beta, \nu_\beta}(s,a,b)\right) \cdot$$

$$\frac{\sum_{b' \in \mathcal{B}} \nu_\beta(s,b') \exp\left(-\frac{1}{\epsilon} \sum_{a \in \mathcal{A}} \mu_k(s,a) Q^{\mu_\beta, \nu_\beta}(s,a,b')\right) \left(Q^{\mu_\beta, \nu_\beta}(s,a,b') - Q^{\mu_\beta, \nu_\beta}(s,a,b)\right)}{\left(\sum_{b' \in \mathcal{B}} \nu_\beta(s,b') \exp\left(-\frac{1}{\epsilon} \sum_{a \in \mathcal{A}} \mu_k(s,a) Q^{\mu_\beta, \nu_\beta}(s,a,b')\right)\right)^2}$$

Now, it is clear:

$$\lim_{\frac{1}{\epsilon} \to 0} \frac{\partial \nu_k(s,b)}{\partial \mu_k(s,a)} = 0 \cdot \frac{\sum_{b' \in \mathcal{B}} \nu_\beta(s,b') \left(Q^{\mu_\beta, \nu_\beta}(s,a,b') - Q^{\mu_\beta, \nu_\beta}(s,a,b)\right)}{\left(\sum_{b' \in \mathcal{B}} \nu_\beta(s,b')\right)^2} = 0$$

### A.4    PROOF OF LEMMA 3

*Proof.* Without loss of generality, we prove the first half that $\mu^*$ is unexploitable with respect to $\nu^*$. We show that $\sum_{a \in \mathcal{A}} \mu^*(s,a) Q^{\mu^*,\nu^*}(s,a,b_1) > \sum_{a \in \mathcal{A}} \mu^*(s,a) Q^{\mu^*,\nu^*}(s,a,b_2)$ leads to a contradiction when $(\mu^*, \nu^*)$ is a Nash equilibrium with full support. By definition, the value at state $s$ is:

$$V^{\mu^*,\nu^*}(s) = \sum_{b \in \mathcal{B}} \nu^*(s,b) \sum_{a \in \mathcal{A}} \mu^*(s,a) Q^{\mu^*,\nu^*}(s,a,b)$$

When $\nu^*(s)$ has nonzero probability at each $b \in \mathcal{B}$, decreasing $\nu^*(s,b_1)$ and increasing $\nu^*(s,b_2)$ should decrease the value for the min-player. Therefore, $\nu^*$ is not a best response against $\mu^*$, which contradicts the NE assumption. $\qquad\square$

### A.5    PROOF OF LEMMA 4

*Proof.* By definition:

$$\sum_{a \in \mathcal{A}} \left( \mu(s,a) - (\mu_k(s,a) + z_a^s) \right)^2$$

$$= \sum_{a \in \mathcal{A}} \left( \mu(s,a) - \left( \mu_k(s,a) + p_a^s + \frac{y}{|\mathcal{A}|} \right) \right)^2$$

$$= \sum_{a \in \mathcal{A}} \left( (\mu(s,a) - (\mu_k(s,a) + p_a^s)) - \frac{y}{|\mathcal{A}|} \right)^2$$

$$= \sum_{a \in \mathcal{A}} (\mu(s,a) - (\mu_k(s,a) + p_a^s))^2 + \sum_{a \in \mathcal{A}} \left( \frac{y}{|\mathcal{A}|} \right)^2 - \frac{2y}{|\mathcal{A}|} \sum_{a \in \mathcal{A}} (\mu(s,a) - (\mu_k(s,a) + p_a^s))$$

$$= \sum_{a \in \mathcal{A}} (\mu(s,a) - (\mu_k(s,a) + p_a^s))^2 + \frac{y^2}{|\mathcal{A}|} - \frac{2y}{|\mathcal{A}|} \left( \sum_{a \in \mathcal{A}} \mu(s,a) - \sum_{a \in \mathcal{A}} \mu_k(s,a) + \sum_{a \in \mathcal{A}} z_a^s - \sum_{a \in \mathcal{A}} \frac{\sum_{a \in \mathcal{A}} z_a^s}{|\mathcal{A}|} \right)$$

$$= \sum_{a \in \mathcal{A}} (\mu(s,a) - (\mu_k(s,a) + p_a^s))^2 + \frac{y^2}{|\mathcal{A}|} - \frac{2y}{|\mathcal{A}|} (1 - 1)$$

$$= \sum_{a \in \mathcal{A}} (\mu(s,a) - (\mu_k(s,a) + p_a^s))^2 + \frac{y^2}{|\mathcal{A}|}$$

Therefore:

$$\mu_{k+1}(s) = \operatorname*{arg\,min}_{\mu(s) \in \Delta(\mathcal{A})} \sum_{a \in \mathcal{A}} (\mu(s,a) - (\mu_k(s,a) + z_a^s))^2 = \operatorname*{arg\,min}_{\mu(s) \in \Delta(\mathcal{A})} \sum_{a \in \mathcal{A}} (\mu(s,a) - (\mu_k(s,a) + p_a^s))^2$$

$$\square$$

A.6 POLICY PENALTY BOUND

We use the following lemma to rigorously demonstrate that the indirect policy penalty in CED can bound the distance between the learned policy $\nu_k$ and the behavior policy $\nu_\beta$.

**Lemma 5** (Policy Penalty Bound). *Let $Q_{\max}$ and $Q_{\min}$ be the maximum and minimum values of $Q^{\mu_\beta, \nu_\beta}$ and let $C > 0$ be any threshold. When $\epsilon \geq \frac{Q_{\max} - Q_{\min}}{\log(1+C)}$, it holds that $\|\nu_k(s) - \nu_\beta(s)\|_1 \leq C$ for all $s \in \mathcal{S}$ in the CED algorithm.*

*Proof.* By Lemma 1, we have:

$$\nu_k(s, b) = \frac{\nu_\beta(s, b) \exp\left(-\frac{1}{\epsilon} \sum_{a \in \mathcal{A}} \mu_k(s, a) Q^{\mu_\beta, \nu_\beta}(s, a, b)\right)}{\sum_{b' \in \mathcal{B}} \nu_\beta(s, b') \exp\left(-\frac{1}{\epsilon} \sum_{a \in \mathcal{A}} \mu_k(s, a) Q^{\mu_\beta, \nu_\beta}(s, a, b')\right)}$$

Let $t = \frac{\nu_\beta(s, b)}{\nu_k(s, b)} = \sum_{b' \in \mathcal{B}} \nu_\beta(s, b') \exp\left(\frac{1}{\epsilon} \sum_{a \in \mathcal{A}} \mu_k(s, a) \left(Q^{\mu_\beta, \nu_\beta}(s, a, b) - Q^{\mu_\beta, \nu_\beta}(s, a, b')\right)\right)$.

By definition of $Q_{\max}$ and $Q_{\min}$, we have:

$$Q_{\min} - Q_{\max} \leq Q^{\mu_\beta, \nu_\beta}(s, a, b) - Q^{\mu_\beta, \nu_\beta}(s, a, b') \leq Q_{\max} - Q_{\min}$$

Since $\sum_{a \in \mathcal{A}} \mu_k(s, a) = 1$, we have:

$$\frac{Q_{\min} - Q_{\max}}{\epsilon} \leq \frac{1}{\epsilon} \sum_{a \in \mathcal{A}} \mu_k(s, a) \left(Q^{\mu_\beta, \nu_\beta}(s, a, b) - Q^{\mu_\beta, \nu_\beta}(s, a, b')\right) \leq \frac{Q_{\max} - Q_{\min}}{\epsilon}$$

Since $\sum_{b' \in \mathcal{B}} \nu_\beta(s, b') = 1$, we further have:

$$\exp\left(\frac{Q_{\min} - Q_{\max}}{\epsilon}\right) \leq t \leq \exp\left(\frac{Q_{\max} - Q_{\min}}{\epsilon}\right)$$

Since $\epsilon \geq \frac{Q_{\max} - Q_{\min}}{\log(1+C)}$, it holds that $\exp\left(\frac{Q_{\max} - Q_{\min}}{\epsilon}\right) \leq 1 + C$. Therefore, $t \leq 1 + C$.

When $C \geq 1$, it is clear that $\exp\left(\frac{Q_{\min} - Q_{\max}}{\epsilon}\right) \geq 1 - C$. When $0 < C < 1$, we have:

$$\epsilon \geq \frac{Q_{\max} - Q_{\min}}{\log(1 + C)} \geq \frac{Q_{\max} - Q_{\min}}{-\log(1 - C)} = \frac{Q_{\min} - Q_{\max}}{\log(1 - C)}$$

It is also clear that $\exp\left(\frac{Q_{\min} - Q_{\max}}{\epsilon}\right) \geq 1 - C$. Therefore, $t \geq 1 - C$.

Since $|\nu_k(s, b) - \nu_\beta(s, b)| = |\nu_k(s, b)(1 - t)| \leq \nu_k(s, b) |1 - t|$, we have:

$$\|\nu_k(s) - \nu_\beta(s)\|_1 \leq \sum_{b \in \mathcal{B}} \nu_k(s, b) |1 - t| = |1 - t| \leq C$$

$\square$

## B    TEST ENVIRONMENTS

### B.1    TREE-FORM GAME

We use a tree-form game $\mathcal{T}$ as a test environment for both CED and ED algorithms. Figure 5 is an illustration of $\mathcal{T}$, which consists of three decision points with payoff matrices $\mathcal{M}_1$, $\mathcal{M}_2$, and $\mathcal{M}_3$, respectively. $\mathcal{T}$ starts with Stage 1 ($\mathcal{M}_1$) and enters Stage 2 ($\mathcal{M}_2$) or Stage 3 ($\mathcal{M}_3$) conditioned on previous actions. If both use the same action 0 or 1, $\mathcal{T}$ enters Stage 2. Otherwise, $\mathcal{T}$ enters Stage 3.

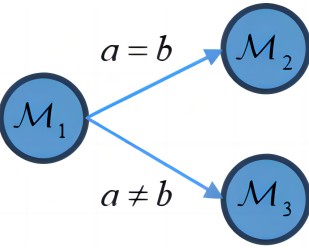

Figure 5: Tree-Form Game

### B.2    SOCCER GAME

We use a two-player zero-sum soccer game as the test environment for infinite-horizon MGs. Figure 6 is an illustration of the game. The two players are marked with A and B. The player who keeps the ball is marked with a cycle. Each player can choose an action from "up", "down", "left", "right", and "stay" at each time step. If the two players collide after the simultaneous move, then the ball possession exchanges. When the ball carrier moves into the opponent's goal, the game terminates. The winning player receives a reward of $+100$ and the opponent receives a reward of $-100$. The initial state distribution $\rho_0$ is set to be uniform, and the discount factor $\gamma$ is set to be $0.95$.

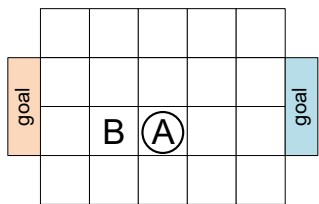

Figure 6: Soccer Game

## C    FURTHER EXPLANATIONS

### C.1    INTUITION FOR THE CONSTRUCTION OF CED

Recall that the NE strategy $\mu^*$ for the max-player always satisfy $\mu^* = \arg\max_\mu \left\{ \min_\nu u(\mu, \nu) \right\}$. The idea of ED (Algorithm 1) is to update $\mu$ along the gradient of $\min_\nu u(\mu, \nu)$. However, this gradient may not exist since $\mathrm{br}(\mu) := \arg\min_\nu u(\mu, \nu)$ may have multiple solutions. Therefore, by fixing an arbitrary $\nu' \in \mathrm{br}(\mu)$, a generalized gradient $\frac{\partial u(\mu, \nu')}{\partial \mu} \in \partial \min_\nu u(\mu, \nu)$ is used instead. As a result, the max-player policy $\mu$ can "converge" to a local Nash equilibrium (see Lockhart et al. (2019)).

For CED (Algorithm 2), since the computation of $\nu$ is under divergence regularization (indirect policy constraint), it is uniquely determined by $\mu$ but is no longer an exact best response to $\mu$. Therefore, the update of $\mu$ does not follow a gradient induced by best response and cannot converge to the NE strategy. However, as long as the limit point is interior in the constrained policy set, we can use

the projected update formula in Lemma 4 to prove that $\mu$ has the same value for all actions at any given state $s \in \mathcal{S}$. Therefore, the min-player policy $\nu$ satisfies the property of mixed-strategy NE, i.e., being unexploitable with respect to its opponent. The NE policies in our matrix/tree-form game experiments in Section 6 are explicit examples.

Note that ED itself does not have this property because the learned policy is unstable around a local optimum of the minimax problem. From the perspective of offline RL, the policy constraints in CED can also mitigate the problem of encountering out-of-distribution states and actions.

### C.2 DISCUSSION ON THE PERFORMANCE OF CED

Please note that our current experimental results are based on relatively small-scale games (like soccer game) and the tabular representation of policies. Actually, there is a realistic reason that makes it a rather difficult task to provide an exact evaluation for CED in games with a large scale. Note that the metric NashConv is based on the computation of worst-case utility, which requires the best response of each player against the opponent policy. When the state representation is complex, we cannot avoid using deep reinforcement learning to approximately compute the best response. As a result, the computed value of NashConv is affected by the choice of the algorithm for evaluation and can deviate from the true value itself. In simpler games like the soccer game, however, this value can be exactly computed through tabular-form dynamic programming, and it is practical to generate the learning curves for comparison purposes.

For large-scale games, where the uniform coverage assumption is not guaranteed, the practical performance of CED can depend on a variety of aspects. We assume that the performance metric of NashConv can be exactly computed. Then, the influential factors can be the data coverage itself, the data quality (the closeness of the behavior policy to Nash equilibrium), the hyperparameters of CED (including the specific policy constraint measure $D(\cdot, \cdot)$ for $\mu$), and the network architecture for state value representation. Based on our existing results and observations, we can provide more information about how these factors affect the performance of CED.

In our experiment for non-uniform coverage, at each state, an action for each player (along with the subsequent states) is directly removed from the dataset of the uniformly random behavior policy. Both data coverage and data quality are poor, which could be the primary reason for not learning a policy close to Nash equilibrium. Besides, we only use the simplest Euclidean distance in the direct policy constraint on $\mu$ and do not employ neural networks. A well-tuned policy constraint measure and a well-designed network architecture for the specific problem can help improve the performance of CED in large-scale games with non-uniform data coverage.

Specifically, Theorem 2 requires the converged max-player policy $\mu$ to be an interior point of the constrained policy set. If the policy constraint measure on $\mu$ is well-tuned, this condition can be better satisfied, and the ultimate policy could also have a smaller NashConv gap. On the other hand, since neural networks may generalize the existing transitions in the dataset to the unknown ones, the performance can be better if CED employs an appropriate network architecture designed for the specific game. Also note that some existing work has pointed out that training value networks using classifications rather than regressions may significantly improve the performance of DRL algorithms in non-stationary environments (see Farebrother et al. (2024)). This technique could also be employed to improve the performance of CED in large-scale games.

## D PARAMETER SELECTION DETAILS

With respect to the learning rate $\alpha$, Theorem 1 provides a guideline that it should be sufficiently small. However, an overly small $\alpha$ will slow down the speed of convergence, as is shown in Figure 3 (mid). Therefore, there is a trade-off about the selection of $\alpha$. For the soccer game, this hyperparameter is not sensitive as long as it is smaller than the threshold of $10^{-5}$.

With respect to the policy penalty parameter $\epsilon$, Theorem 1 also provides a guideline that it should not be overly small, as is verified in Figure 3 (right). However, it is also risky to set an overly large $\epsilon$ because the interior point condition in Theorem 2 is implicitly affected by the policy constraint on the min-player policy $\nu$. For the soccer game, this hyperparameter is supposed to be within $[10^{-3}, 10^{-1}]$.

