# OpenReview forum: "Constrained Exploitability Descent: Finding Mixed-Strategy Nash Equilibrium by Offline Reinforcement Learning"
_ICLR.cc/2025/Conference — Submitted to ICLR 2025_

### Official Review · Reviewer_EzXC · 2024-10-20

**Soundness:** 3
**Presentation:** 3
**Contribution:** 3
**Rating:** 6
**Confidence:** 3

**Summary:**

This paper introduces Constrained Exploitability Descent (CED), a novel model-free offline reinforcement learning algorithm designed to find mixed-strategy Nash equilibria in adversarial Markov games. The key contributions of the paper are:

1. The authors propose CED, which combines game-theoretic approaches with policy constraint methods from offline RL to address the challenge of learning in adversarial settings, where the optimal policy can be a mixed-strategy Nash equilibrium.
2. In contrast to previously proposed model-based methods, CED no longer relies on generalized gradient computation.
3. The authors provide experimental validation of CED in matrix games, a tree-form game, and an infinite-horizon soccer game. The results demonstrate that CED can effectively find optimal min-player policies when practical offline data guarantees uniform coverage and achieves significantly lower NashConv compared to existing pessimism-based methods.

**Strengths:**

The primary strengths of this paper can be summarized in the following two aspects:

1. The authors present a novel model-free offline reinforcement learning algorithm that improves upon previously proposed methods for solving adversarial Markov games.
2. The paper provides both rigorous theoretical analysis and thorough empirical validation.

**Weaknesses:**

Here are the major weaknesses from my perspective:

1. While I acknowledge the authors' contribution in improving the performance of Exploitability Descent, they only compare their method to one RL algorithm for solving adversarial Markov games. This limits the contribution, as the authors have not compared their method to other previously proposed approaches, both from a theoretical and empirical standpoint.
2. In the experimental implementation, the authors only evaluate their algorithm on games with relatively simple structures. This raises uncertainty about whether the proposed algorithm achieves significant improvement over existing methods in more complex, practical tasks.
3. The discussion on the coverage condition is insufficient. The assumption of uniform coverage seems quite strong in the offline RL literature, and the authors have not explored how their algorithm performs under other, less restrictive coverage conditions.

**Questions:**

Based on the discussion of the paper's strengths and weaknesses, I have the following questions for the authors:

1. Is the uniform coverage assumption essential for the analysis? Could the proposed algorithm also work under different coverage conditions?
2. Can the proposed method be extended to other game-theoretic settings, such as multi-agent general-sum games?
3. Does CED outperform other algorithms for adversarial zero-sum Markov games beyond Exploitability Descent (ED)?

---

> ### Author Response · Authors · 2024-11-23
>
> Thank you for reviewing this paper and providing valuable comments. Here, we reply to your questions and concerns about the paper.
>
> **[Questions]**
>
> _**Question 1**_:
>
> > Is the uniform coverage assumption essential for the analysis?
>
> Yes, the uniform coverage assumption is theoretically essential. Actually, [1] provides a counterexample where the exact NE becomes impossible to learn when the offline dataset does not guarantee uniform coverage. Besides, the global optimality of ED also requires that all states should be visited with non-zero probability. Under offline settings, this requirement also implies uniform coverage.
>
> > Could the proposed algorithm also work under different coverage conditions?
>
> Yes, we actually provide some experimental results to show that CED can work under different coverage conditions. The last soccer game experiment in Section 6.3 shows that CED can gradually improve behavior policy under different non-uniform coverages and eventually obtain a policy with fair win rates.
>
> _**Question 2**_:
>
> > Can the proposed method be extended to other game-theoretic settings, such as multi-agent general-sum games?
>
> Yes, further extensions to other game-theoretic settings are highly possible, and we also mention that in Section 7. Exploitability Descent (ED) itself is capable of solving imperfect-information games (IIGs). Since CED is constructed based on this game-theoretic approach, it is possible to use CED as an offline IIG solver by replacing the state and value with information state and counterfactual value. However, how to estimate counterfactual value based on the current policy and offline game data remains an open problem. In order to guarantee a stable performance, further theoretical analysis is still required. For multi-agent general-sum games, finding Nash equilibrium is a PPAD-hard problem, and ED itself is not guaranteed to find NE in this scenario. While CED may not be directly applicable as well, the idea of policy constraint can be generally employed to extend more online equilibrium-learning algorithms (even for multiplayer games) to offline settings.
>
> _**Question 3**_:
>
> > Does CED outperform other algorithms for adversarial zero-sum Markov games beyond Exploitability Descent (ED)?
>
> While online algorithms like ED can solve adversarial zero-sum Markov games, they cannot directly learn from a set of offline data. The basic function of policy constraint in CED is to guarantee that it can work offline and improve the behavior policy. While CED has an advantage in the online comparison with ED, we do not expect it to outperform other algorithms in an online setting. In the soccer game, we compare CED with a recently proposed pessimism-based method [2], which is among the few offline game solvers with an NE guarantee in infinite-horizon adversarial games. CED demonstrates a clear advantage that it can find NE at a relatively high precision under a finite set of offline data.
>
> **References**:
>
> [1] Qiwen Cui and Simon S Du. When are offline two-player zero-sum Markov games solvable? Advances in Neural Information Processing Systems, 35:25779–25791, 2022.
>
> [2] Yuling Yan, Gen Li, Yuxin Chen, and Jianqing Fan. Model-based reinforcement learning for offline zero-sum Markov games. Operations Research, 2024.
>
> Thanks again for your comments. We are looking forward to having further discussions with you.

---

> ### Comment · Reviewer_EzXC · 2024-11-25
> **Thank You**
>
> Thanks the authors for their response, and I will keep my score (which is positive).

---

### Official Review · Reviewer_sPGt · 2024-10-22

**Soundness:** 1
**Presentation:** 2
**Contribution:** 1
**Rating:** 3
**Confidence:** 5

**Summary:**

This paper studies the problem of offline learning Nash Equilibrium in model-free deterministic two-player zero-sum Markov games. It combines the techniques from exploitability descent and policy constraints in offline learning setting and proposed an algorithm CED. The authors further showed that, under suggested assumptions, the proposed algorithm might converge to a NE. This paper provided some numerical results to support their theorems.

**Strengths:**

1. The proposed algorithm achieves last-iterate convergence under suggested assumptions.
2. Numerical results and graphs are provided to support the proposed theorems.

**Weaknesses:**

1. The theoretical novelty of this paper is limited.
2. Theorem 1 only provides asymptotic convergence analysis, which is sharp contrast even with the direct previous work [1].
3. The proof of theorem 1 relies on the assumption that $\frac{1}{\epsilon} \to 0$. However, in the provided numerical experiments, all $\epsilon$ are set to be small (e.g. $1, 0.1, 10^{-3}, 10^{-4}$).
4. Theorem 2 relies on the assumption that the final stationary point $\bar{\mu}(s)$ lies in the interior of the feasibility set $\Pi(s)$. However, it seems to be a relative strong assumption, for example, it requires the Markov game to have a NE with full support. The paper claims that previous work [1] posed the same assumption but I couldn't find any reference in [1] regarding this assumption. I would be grateful if the authors can specifically indicate where the assumption is specifically stated during their rebuttal.
5. This work establish the convergence to NE by stating that $Q^{\mu, \hat{\nu}} \approx Q^{\mu_{\beta}, \hat{\nu}} \approx Q^{\hat{\mu}, \hat{\nu}}$ and reach the conclusion that $(\hat{\mu}, \hat{\nu})$ forms a NE (386-387). I believe that more rigorous mathmatical statements are required to reach this conclusion, for example, can the error bound  $||\mu_{\beta} - \hat{\mu}||$ be provided?
6. Behavior policies are not provided in numerical results.
7. In line 203 the regularizing term should be $+ \epsilon D(\pi(s), \pi_{\beta}(s))$ instead of negative.

[1] Edward Lockhart, Marc Lanctot, Julien Pe ́rolat, Jean-Baptiste Lespiau, Dustin Morrill, Finbarr Timbers, and Karl Tuyls. Computing approximate equilibria in sequential adversarial games by exploitability descent. In Proceedings of the 28th International Joint Conference on Artificial Intelligence, pp. 464–470, 2019.

**Questions:**

1. In the last line of  proof of lemma 4, I don't see why the term $\frac{y^2}{|A|}$ disappears.
2. In lemma 4 it is stated that $(\bar{\mu}(s, a) + p^s_a)_{a \in A}$ stays in the probability simplex. However, it is not clear to me why this is the case. I would appreciate if the authors can provide some insights during their rebuttal.

---

> ### Author Response · Authors · 2024-11-23
>
> Thank you for reviewing this paper and providing valuable comments. Here, we reply to your questions and concerns about the paper.
>
> **[Weaknesses]**
>
> _**Weakness 1**_:
>
> > The theoretical novelty of this paper is limited.
>
> Actually, we feel safe to say our theoretical analysis is novel. While CED resembles ED [1], their learning behaviors are quite different. We believe the authors of ED have not expected that **the algorithm can be employed to optimize (opponent) min-player's policy**, which they simply use as an auxiliary policy when optimizing (self) max-player's policy. This finding was even surprising for us. Orginally, we designed CED just in hope of optimizing the max-player under policy constraints. However, the empirical results were in contrast to our expectation, showing that the min-player rather than the max-player can converge to an NE. We establish corresponding theoretical analysis to explain the finding through this paper. At the same time, **the introduction of policy constraints makes the algorithm no longer rely on a generalized gradient and mitigates action distributional shift under offline settings**. These are the two core advantages of CED over ED.
>
> _**Weakness 2**_:
>
> > Theorem 1 only provides asymptotic convergence analysis, which is sharp contrast even with the direct previous work [1].
>
> Please note that Theorem 1 directly proves last-iterate convergence, while [1] only provides rigorous guarantees for the best iterate as it relies on a generalized gradient. Besides, the convergence analysis in [1] is based on several existing results on min-max optimization and regret minimization. Since these results are not applicable to our CED method with policy constraints, we instead use common mathematical tools to prove asymptotic last-iterate convergence for CED.
>
> _**Weakness 3**_:
>
> > The proof of theorem 1 relies on the assumption that $\frac{1}{\epsilon} \to 0$. However, in the provided numerical experiments, all $\epsilon$ are set to be small (e.g. $1, 0.1, 10^{-3}, 10^{-4}$).
>
> Actually, the condition $\frac{1}{\epsilon} \to 0$ is for convenience of analysis. It only satisfies a sufficient condition to use large $\epsilon$. In practice, the convergence is not affected as long as $\epsilon$ is not overly small.
>
> _**Weakness 4**_:
>
> > The paper claims that previous work [1] posed the same assumption but I couldn't find any reference in [1] regarding this assumption.
>
> We admit that [1] does not necessarily require interior point assumption and have clarified it in the current submission. The assumption that [1] directly requires is that all states should be visited with non-zero probability. Under offline settings, this assumption instantly implies uniform coverage on states. When the uniform coverage is satisfied, our interior point assumption can be viewed as a sufficient condition for the assumption in [1] to hold.
>
> _**Weakness 5**_:
>
> > I believe that more rigorous mathematical statements are required to reach this conclusion, for example, can the error bound $\left\\| {\mu } _ {\beta }-\hat{\mu } \right\\|$ be provided?
>
> Yes, we can provide more detailed analysis. For example, let $Q _ {\max}$ ($Q _ {\min}$) be the maximum (minimum) Q value under behavior policy $(\mu _ {\beta},\nu _ {\beta})$. Given a threshold $C>0$, we can prove that $\forall s\in\mathcal{S},\left\\| {{\mu } _ {\beta }}(s)-\hat{\mu }(s) \right\\|\le C$ when $\epsilon\geq\frac{Q _ {\max}-Q _ {\min}}{\log \left( 1+C \right)}$. This provides an explicit error bound for $\left\\| {\mu } _ {\beta }-\hat{\mu } \right\\|$. In the current submission, we have further clarified how the conclusion is reached and provided the error bound using a new lemma (Appendix B.6).
>
> _**Weakness 6**_:
>
> > Behavior policies are not provided in numerical results.
>
> For the experiments under uniform coverage, the behavior policies are all uniform policies. In Section 6.1, we actually specify that the behavior policy $(\mu _ {\beta},\nu _ {\beta})$ is uniform. In Section 6.3, we clarify that under uniform coverage, the dataset uses the minimum amount of samples. This also implies the behavior policy is uniform. For the non-uniform coverage experiment, the behavior policy is determined by how the transitions are removed from the dataset. Since the removal involves randomness and results in an imbalanced dataset, the behavior policy cannot be explicitly described.
>
> _**Weakness 7**_:
>
> > In line 203, the regularizing term should be $+\epsilon D(\pi(s) ,{{\pi } _ {\beta }}(s))$ instead of negative.
>
> Thank you for your careful reading. Actually, the sign should be kept negative here, but the operators outside should be $\arg\max$ rather than $\arg\min$ for the two formulas. We are sorry for the typo and have revised it in the current submission.

---

> > ### Comment · Reviewer_sPGt · 2024-11-26
> >
> > I thank the author for detailed response. However, some of my main concerns of this paper remain unresolved.
> >
> > > Weakness 2:
> >
> > It is not impossible to provide non-asymptotic rate for last-iterate convergence. In fact, many works have established last-iterate convergence rate for other settings (e.g. [1]). If the authors believe that there is a significant theoretical obstacle that prevents achieving non-asymptotic convergence rate, such justification should be included in the paper.
> >
> > > Weakness 3:
> >
> > The experiment results should validate the theoretical findings. If $\frac{1}{\epsilon} \to 0$ serves as a sufficient condition, I believe experiments that conform with this condition should be provided.
> >
> > > Weakness 4:
> >
> > Other than the uniform coverage assumption, it seems that the paper further requires the existence of a full-supported Nash in order to establish convergence to Nash Equilibrium. This assumption becomes very strong as the size of the game goes large and no justification of this assumption is provided.
> >
> >
> > Since some of my major concerns remain unresolved, I would like to keep my score the same.
> >
> >
> >
> > [1] Chung-Wei Lee, Haipeng Luo, Chen-Yu Wei, and Mengxiao Zhang. Linear last-iterate convergence for matrix games and stochastic games. arXiv preprint arXiv:2006.09517, 2020.

---

> ### Author Response · Authors · 2024-11-23
>
> **[Questions]**
>
> _**Question 1**_:
>
> > In the last line of proof of lemma 4, I don't see why the term $\frac{y^2}{\left\|\mathcal{A}\right\|}$ disappears.
>
> Please note that the computation of $y$ does not involve $\mu$. Therefore, the two $\arg\min$ in the last line result in the same $\mu_{k+1}$.
>
> _**Question 2**_:
>
> > In lemma 4 it is stated that ${\left( \mu (s,a)+p _ {a} ^ {s} \right)} _ {a\in \mathcal{A}}$ stays in the probability simplex. However, it is not clear to me why this is the case.
>
> Please note that we only say that ${\left( \mu (s,a)+p _ {a} ^ {s} \right)} _ {a\in \mathcal{A}}$ stays in the hyperplane that the probability simplex belongs to (i.e., $\sum\nolimits _ {a\in \mathcal{A}}{x _ a}=1$). This is actually what we need in the proof of Theorem 2.
>
> **Reference**:
>
> [1] Edward Lockhart, Marc Lanctot, Julien Pérolat, Jean-Baptiste Lespiau, Dustin Morrill, Finbarr Timbers, and Karl Tuyls. Computing approximate equilibria in sequential adversarial games by exploitability descent. In Proceedings of the 28th International Joint Conference on Artificial Intelligence, pp. 464–470, 2019.
>
> Thanks again for your comments. We are looking forward to having further discussions with you.

---

### Official Review · Reviewer_iyxn · 2024-10-24

**Soundness:** 3
**Presentation:** 3
**Contribution:** 3
**Rating:** 6
**Confidence:** 2

**Summary:**

The paper presents Constrained Exploitability Descent (CED), a novel model-free offline RL algorithm aimed at finding mixed-strategy Nash equilibria in two-player zero-sum Markov games. CED combines game-theoretic approaches with policy constraint methods from offline RL to find mixed-strategy NEs. The authors also provide theoretical results, showing that CED converges to an unexploitable policy under certain conditions. Finally, experiments in matrix games, a tree-form game, and a soccer game validate that CED outperforms existing methods in achieving lower NashConv. The algorithm is notable for its stability and effectiveness, even with limited offline data.

**Strengths:**

1.	The introduction of Constrained Exploitability Descent (CED) is an innovative contribution to offline reinforcement learning in adversarial two-player zero-sum Markov games. By combining the concepts of Exploitability Descent (ED) with policy constraints typically seen in offline RL, the paper introduces a new approach to handling distributional shifts in a competitive setting.

2.	The paper provides solid theoretical results, proving that for CED converges to a stationary point in deterministic two-player zero-sum Markov games under uniform data coverage assumptions.

3.	The empirical results are thorough and well-presented. The experiments span various scenarios, including matrix games, tree-form games, and a soccer game, demonstrating the practical applicability and effectiveness of CED in different settings.

**Weaknesses:**

1.	The convergence proof relies on the assumption of uniform data coverage, which may not always hold in real-world offline RL settings.

2.	The paper does not provide detailed insights into the computational complexity or scalability of CED, especially in environments with larger state or action spaces.

**Questions:**

As I’m not an expert in this area, I have listed my questions below. If there are any mistakes, please feel free to correct me.

Q1: The convergence proof of CED relies on uniform data coverage. Could you elaborate on how the algorithm performs when this assumption is not met, especially in cases of highly imbalanced or sparse datasets? Are there mechanisms within CED to mitigate these issues, or would it require additional modifications?

Q2: Could you provide more insights into the scalability of CED in larger games with high-dimensional state-action spaces or more complex multi-agent interactions? Or have you tested its computational efficiency or considered potential bottlenecks when scaling up to larger environments?

---

> ### Author Response · Authors · 2024-11-23
>
> Thank you for reviewing this paper and providing valuable comments. Here, we reply to your questions and concerns about the paper.
>
> **[Questions]**
>
> _**Question 1**_:
>
> > The convergence proof of CED relies on uniform data coverage. Could you elaborate on how the algorithm performs when this assumption is not met, especially in cases of highly imbalanced or sparse datasets? Are there mechanisms within CED to mitigate these issues, or would it require additional modifications?
>
> Actually, we feel safe to say our experiment for non-uniform data coverage (Figure 4) has already been conducted in an imbalanced dataset. At each state, a random action for each player (along with the subsequent state) is directly removed from the dataset of uniformly random behavior policy. As a result, both data coverage and data quality are poor, which could be the primary reason for not learning a policy with sufficiently small NashConv. Besides, we only use the simplest Euclidean distance in the direct policy constraint on $\mu$ and do not employ neural networks. A well-designed policy constraint measure or network architecture for the specific problem can help improve the performance of CED in practical offline games under non-uniform data coverage.
>
> Specifically, Theorem 2 requires the converged max-player policy $\mu$ to be an interior point of the constrained policy set. If the policy constraint measure on $\mu$ is well-tuned, this condition can be better satisfied, and the learned policy could have a smaller NashConv gap. Besides, since neural networks may generalize the existing transitions in the dataset to the unknown ones, the performance can be better if CED employs an appropriate network architecture for the specific game. Even without further modifications, the basic policy constraint mechanism already mitigates action distributional shift (an inherent challenge for offline RL) and thus allows CED to steadily improve the performance starting from the behavior policy.
>
> _**Question 2**_:
>
> > Could you provide more insights into the scalability of CED in larger games with high-dimensional state-action spaces or more complex multi-agent interactions? Or have you tested its computational efficiency or considered potential bottlenecks when scaling up to larger environments?
>
> For large-scale games, the practical performance of CED can depend on a variety of aspects: the data coverage itself, the data quality (the closeness of the behavior policy to Nash equilibrium), the hyperparameters of CED (including the specific policy constraint measure $D(\cdot,\cdot)$ for $\mu$), and the network architecture for state value representation. However, there is a realistic reason that makes it a rather complex task to provide an exact evaluation for CED in games with a large scale. Note that the metric NashConv is based on the computation of worst-case utility, which requires the best response of each player under the current adversarial policy. When the state representation is complex, deep reinforcement learning will be employed to approximately compute the best response. As a result, the computed value of NashConv is affected by the choice of the algorithm for evaluation and can deviate from the true value itself. In simpler games like the soccer game, however, this value can be exactly computed through tabular-form dynamic programming, and it is practical for use to generate the learning curves for comparison purposes.
>
> Based on our current experience in large-scale games, one of the potential bottlenecks in the learning process can be the non-stationary environment itself. Since we are dealing with adversarial games, the training can be unstable, especially when it scales up to larger environments. Under offline settings, the non-stationarity persists since the opponent policy keeps changing. Still, a range of RL techniques can be employed to mitigate the problem. For example, training value networks using classifications rather than regressions may significantly improve the performance of DRL algorithms in non-stationary environments (see [1]). This technique could be employed to improve the performance of CED in large-scale games.
>
> **Reference**:
>
> [1] Jesse Farebrother, Jordi Orbay, Quan Vuong, Adrien Ali Taïga, Yevgen Chebotar, Ted Xiao, Alex Irpan, Sergey Levine, Pablo Samuel Castro, Aleksandra Faust, et al. Stop regressing: Training value functions via classification for scalable deep RL. International Conference on Machine Learning, 2024.
>
> Thanks again for your comments. We are looking forward to having further discussions with you.

---

> > ### Comment · Reviewer_iyxn · 2024-11-28
> >
> > Thanks to the authors for their detailed responses.
> >
> > Regarding Q1, the authors mentioned that Figure 4 (right) presents the results under the dataset without uniform coverage. However, I find it confusing why the learned policy from non-uniform coverage achieves a win rate of 91.8% against the behavior policy from non-uniform coverage, while the win rate drops significantly to 45.8% when facing the learned policy from uniform coverage. Could the authors clarify this disparity?

---

> ### Author Response · Authors · 2024-11-28
>
> Thanks for the question. Please note that the behavior policy corresponds to the exact action distribution of the offline data. Through CED, the learned policy will have a significant performance improvement over the behavior policy. Actually, the learned policy from uniform coverage is close to Nash equilibrium and much better than the behavior policy from either uniform or non-uniform coverage. Since the game is two-player zero-sum and symmetric, no policy can achieve an expected win rate over 50% against a Nash equilibrium in theory. The win rate of 45.8% reflects that the learned policy from non-uniform coverage is practically competitive against a near-optimal policy, though itself is not guaranteed to be a Nash equilibrium.

---

> > ### Comment · Reviewer_iyxn · 2024-11-29
> >
> > Thank you for clarifying. That explanation makes sense to me, so I will maintain my score as it is.

---

### Meta-Review · Area_Chair_LD7c · 2024-12-21

**Metareview:**

The paper presents Constrained Exploitability Descent (CED), which is a descent type algorithm on the exploitability (also known as duality gap or Nash gap) in order to find Nash equilibria in two-player zero-sum Markov games. The approach does not seem novel in the sense that Descent type algorithms on the exploitability have been proposed before for normal form games, but in terms of Markov Games the AC believes that the proposed approach is kind of novel and interesting. The reviewers also had mixed reviews, two were slightly positive and one strongly negative. In terms of the technical meat, the AC will agree with the negative reviewer; the AC does not see much novelty in here. There are some other weaknesses mentioned (e.g only asymptotic convergence is shown, however prior works have rates). Overall this is a borderline paper with a slight opinion towards rejection.

**Additional Comments On Reviewer Discussion:**

None of the reviewers changed their opinion during the rebuttal phase.

---

### Decision · Program_Chairs · 2025-01-22

Reject